# The Risk of Hepatitis B Virus Reactivation in Rheumatoid Arthritis Patients Receiving Tocilizumab: A Systematic Review and Meta-Analysis

**DOI:** 10.3390/v16010078

**Published:** 2024-01-03

**Authors:** Ping-Hung Ko, Meng Hsuan Kuo, I-Ting Kao, Chen-Yi Wu, Chih-Wei Tseng, Shih-Chieh Shao

**Affiliations:** 1School of Medicine, Tzuchi University, Hualien 970, Taiwan; dm226859@tzuchi.com.tw; 2Division of Gastroenterology, Department of Internal Medicine, Dalin Tzu Chi Hospital, Buddhist Tzu Chi Medical Foundation, Chia-Yi 622, Taiwan; 3Department of Pharmacy, Dalin Tzu Chi Hospital, Buddhist Tzu Chi Medical Foundation, Chia-Yi 622, Taiwan; df921180@tzuchi.com.tw (I.-T.K.); df547851@tzuchi.com.tw (C.-Y.W.); 4Department of Pharmacy, Keelung Chang Gung Memorial Hospital, Keelung 204, Taiwan; scshao@cgmh.org.tw

**Keywords:** HBV reactivation, hepatitis flare-up, rheumatoid arthritis, tocilizumab

## Abstract

**Highlights:**

This systematic review on RA patients receiving tocilizumab reveals a notable risk of hepatitis B virus reactivation. These findings emphasize the importance of implementing antiviral prophylaxis and monitoring to mitigate hepatic side effects from tocilizumab in RA patients.

**What is the main finding**
The risk of HBV reactivation from tocilizumab in rheumatology patients cannot be ignored.HBsAg^−^/anti-HBc^+^ patients may need long term follow-up policy when using tocilizumab.Anti-HBs may have protection effect in rheumatology patients treating with tocilizumab.

**Abstract:**

Background: Tocilizumab has demonstrated optimal efficacy and safety in patients with rheumatoid arthritis (RA) from clinical trials. However, the risk of hepatitis B virus reactivation (HBVr) in these patients remains uncertain because patients with underlying HBV have been excluded in phase III studies. Methods: Systematical reviews were conducted on PubMed, Embase, and the Cochrane Central Register of Controlled Trials up to 21 February 2023. Random-effects meta-analysis was performed to calculate the pooled incidence of HBV reactivation. Results: We included 0 clinical trials and 11 observational studies with a total of 25 HBsAg^+^ and 322 HBsAg^−^/anti-HBc^+^ RA patients. Among the HBsAg^+^ patients without antiviral prophylaxis, the pooled rate was 69.4% (95% CI, 32.9–91.3), with a median time of 4 months (range, 1–8 months) from tocilizumab initiated. Half of these patients with HBVr experienced hepatitis flare-up but no deaths. HBVr was eliminated with prophylaxis in this population. Among HBsAg^−^/anti-HBc^+^ patients, the pooled incidence of reactivation was 3.3% (95% CI, 1.6–6.7), with a median time of 10 months (range, 2–43 months) from tocilizumab initiated. HBVr was not associated with hepatitis flare-up and death. HBsAg^−^/anti-HBc^+^ patients without anti-HBs antibodies had a significantly higher risk of HBVr (Odds ratio, 12.20; 95% CI, 1.16–128.06). Conclusions: This systematic review indicated that the risk of HBVr in RA patients with anti-HBs^−^, HBsAg^+^, or HBsAg^−^/anti-HBc^+^ cannot be ignored but may be avoided. Clinicians should consider implementing appropriate antiviral prophylaxis and monitoring policies for RA patients to avoid unnecessary hepatic side effects from tocilizumab treatment.

## 1. Introduction

Interleukin-6 (IL-6), a cytokine with immunomodulatory properties, performs a wide range of functions in maintaining homeostasis and influencing the outcomes of infectious, inflammatory, and autoimmune diseases [1,2]. Tocilizumab (TCZ), a monoclonal antibody that specifically targets both soluble and membrane-bound forms of the IL-6 receptor, has been developed as a biologic agent [3]. TCZ has proven effective in the treatment of a variety of rheumatic diseases, including rheumatoid arthritis, juvenile idiopathic arthritis, and giant cell arteritis [2]. Notably, the combination of tocilizumab and corticosteroids has been employed in severe coronavirus disease 2019 (COVID-19) cases to attenuate the cytokine storm and confer, conferring a modest reduction in mortality [4,5].

Because IL-6 inhibits hepatitis B virus (HBV) replication, the possibility of HBV reactivation with tocilizumab administration is of concern [6,7,8]. Several studies have addressed the risk of HBV reactivation in patients with severe COVID-19 receiving TCZ therapy [9,10]. The risk seems low in patients who are HBV-surface-antigen–negative/HBV-core-antibody–positive (HBsAg^−^/anti-HBc^+^), and a short course of antiviral prophylaxis may be a safe option [11]. A study investigating the outcomes of 44 HBsAg^−^/anti-HBc^+^ COVID-19 patients treated with tocilizumab [9] found that 61% of these patients received prophylactic entecavir. During a 1- to 2-month follow-up, only one patient developed detectable HBV DNA. However, long-term treatment requirements in rheumatoid patients differ from those of COVID-19 patients, potentially affecting the risk of reactivation.

Assessing the risk of TCZ-induced HBV reactivation in patients with rheumatoid arthritis remains challenging. Previous TCZ clinical trials excluded patients who tested positive for HBsAg [12,13], and current guidelines recommend antiviral prophylaxis for HBsAg^+^ individuals before initiating immunosuppressive or cytotoxic therapy [14,15,16]. These factors contribute to limited reporting of HBV reactivation rates among HBsAg^+^ patients. The reported HBV reactivation rates among HBsAg^−^/anti-HBc^+^ patients are inconsistent, ranging from 0% to 11.1% [3,17,18,19,20,21,22,23,24,25], indicates a risk spectrum from low to high according to individual studies. This variability complicates clinical decision-making regarding antiviral prevention or close monitoring. Although a recent meta-analysis reported a pooled reactivation rate of 0.0% (*I*^2^, 0%; P, 0.43) [11], it is important to note that this analysis was based on only 4 studies and a search just through 2021. Therefore, an updated search is needed to include a broader range of studies to strengthen the existing evidence. Additionally, studies have shown that the presence of HBV surface antibody (anti-HBs) is associated with a reduced risk of reactivation in HBsAg^−^/anti-HBc^+^ patients undergoing chemotherapy and biologic agent treatment [18,26,27,28]. However, no study has specifically investigated the effect of anti-HBs on the HBV reactivation rate in HBsAg^−^/anti-HBc^+^ patients receiving tocilizumab. Considering the inconsistent findings and the recent publication of an updated study [17], we conducted a meta-analysis study to investigate the risk of HBV reactivation associated with tocilizumab.

This systematic review and meta-analysis aims to assess the risk of HBV reactivation associated with the use of TCZ in rheumatoid patients, including both HBsAg^+^ and HBsAg^−^/anti-HBc^+^ individuals. Additionally, we aim to determine whether the presence of anti-HBs influences the risk of HBV reactivation in HBsAg^−^/anti-HBc^+^ patients. Furthermore, we recorded the patterns of HBV reactivation throughout the analysis.

## 2. Materials and Methods

Study methods adhered to the Preferred Reporting Items for Systematic Reviews and Meta-Analyses statement (PRISMA) guidelines [29] (Appendix A). Study selection, data extraction, and risk of bias assessments were conducted independently by two authors (ITK and CYW). Disagreements were resolved with the assistance of two senior authors (MHK and CWT). The protocol was pre-registered on INPLASY (INPLASY 202360029).

### 2.1. Search Strategy

On 21 February 2023, we conducted a comprehensive search of PubMed, Embase, and Cochrane Central Register of Controlled Trials to identify relevant published studies focusing on HBV reactivation in patients with rheumatologic disease undergoing tocilizumab treatment. Our search strategy used appropriate MeSH or Emtree terms related to HBV reactivation and tocilizumab, employing free-text search methods. Additionally, we manually examined the reference lists of relevant original studies and reviews to identify any additional articles of interest. Language restrictions were not imposed on the search. For a more detailed overview of the search strategy, please refer to Appendix A.

### 2.2. Study Selection

Following the removal of duplicate records across databases, two reviewers (ITK and CYW) independently assessed the eligibility of studies based on the following inclusion criteria: (1) observational studies (case–control, cross-sectional, or cohort) or randomized trials; (2) inclusion of patients with rheumatologic disease who were either HBsAg^+^ or HBsAg^−^/anti-HBc^+^ and receiving tocilizumab; and (3) provision of data on HBV reactivation using virological and/or biochemical definitions.

The primary endpoint of our research was the reactivation of HBV, as determined by virological and serological assessments. Data extraction focused on elevated HBV DNA levels or the presence of detectable DNA in HBsAg^+^ patients. For patients who were HBsAg^−^/anti-HBc^+^, HBV reactivation was characterized as either a transition of HBV DNA levels from undetectable to detectable or the re-emergence of HBsAg. Hepatitis was clinically defined by increased liver enzyme levels, with a particular emphasis on significant elevations in aspartate aminotransferase (AST) or alanine aminotransferase (ALT), typically 2–3 times above the baseline or exceeding 100–120 U/L, based on the condition of HBV DNA elevation.

We excluded case reports, case series, reviews, meta-analyses, conference papers, animal models, studies with case numbers less than 5 [30], those lacking HBV status reporting, and those having overlapping populations.

### 2.3. Data Extraction and Risk of Bias Assessment

The following information was extracted from the included studies: the first author’s name, publication year, country, setting, study design, treatment regimens, number of patients, age, HBV status (HBsAg, anti-HBc^+^, and anti-HBs), antiviral prophylactics used, definition of HBV reactivation, definition of hepatitis flare-up, follow-up period, and statistical data regarding the prevalence of HBV reactivation and hepatitis flare-up. We extracted only the hepatitis flare-up events that were specifically linked to HBV reactivation resulting from TCZ administration. The patterns of HBV reactivation were also extracted and summarized. In cases where relevant data were missing, we reached out to the authors for clarification. Any disagreements were resolved through discussions. Following the guidelines of the American Gastroenterological Association [16], the risk of HBV reactivation is divided into low risk (if the rate of HBV reactivation is <1%), moderate risk (if the risk of reactivation is between 1–10%), and high risk (if the risk of reactivation is ≥10%).

The risk of bias in the clinical trials and observational studies included in our analysis was assessed using and Cochrane risk-of-bias assessment tool 2.0 and the Newcastle-Ottawa Scale (NOS), respectively [31]. For NOS, a score of 9 stars indicated a low risk of bias, while scores of 7 or 8 stars denoted a moderate risk of bias. Studies that scored 6 stars or less were considered to have a high risk of bias. The NOS has demonstrated comparable reliability to other tools used for assessing risk of bias and has been widely employed to evaluate the methodological quality of observational studies in previous systematic reviews and meta-analyses [32].

### 2.4. Statistical Analysis

We conducted a random-effects meta-analysis of single proportions to estimate the pooled rate of HBV reactivation among rheumatologic disease patients with HBsAg^−^/anti-HBc^+^ receiving tocilizumab. Statistical heterogeneity across the included studies was quantified using the *I*^2^ statistic, with substantial heterogeneity defined as *I*^2^ > 50% [33]. Data also were analyzed within the following subgroups: the definition of HBV reactivation (HBV DNA reappearance or elevation, HBsAg seroreversion, and both criteria), study region (Asian and non-Asian areas), risk of bias (moderate and high), study design (prospective and retrospective), and anti-HBs status. Publication bias was evaluated by assessing funnel plot asymmetry for meta-analyses of outcomes that included ≥10 studies [34]. To assess the robustness of the results of our main analyses, we conducted a sensitivity analysis using the leave-one-out meta-analysis. Two-side *p* < 0.05 was considered statistically significant, and all analyses were performed using Comprehensive Meta-Analysis software (version 4.0, Biostat, Englewood, NJ, USA) and Review Manager Version 5.3 (Cochrane Collaboration, 2020).

## 3. Results

### 3.1. Literature Search and Study Selection

Our search identified 151 relevant published studies. Following the exclusion of 31 duplicate studies, the remaining 120 studies were subjected to screening. Subsequent assessment of titles and abstracts led to the evaluation of 43 full-text studies for eligibility. However, 32 of them did not meet the inclusion criteria and were excluded. Of the excluded studies, seven had case numbers less than five, eighteen lacked HBV status reporting, and seven had overlapping populations. The flow diagram shown in Figure 1 illustrates the process used to select studies for this investigation.

### 3.2. Characteristics of Included Studies

This meta-analysis included 0 clinical trials and 11 observational studies involving a total of 347 patients (25 HBsAg^+^; 322 HBsAg^−^/anti-HBc^+^) with rheumatoid arthritis who were treated with TCZ [3,17,18,19,20,21,22,23,24,25,35]. Of these studies, nine had a retrospective design [3,17,18,19,20,22,24,25,35] and two were prospective [21,23]. The HBV reactivation rate was reported for both HBsAg^+^ and HBsAg^−^/anti-HBc^+^ patients in two of the studies [17,23], for HBsAg^+^ patients only in one study [35], and in HBsAg^−^/anti-HBc^+^ patients who received no antiviral prophylaxis in eight studies [3,18,19,20,21,22,24,25]. In summary, this analysis included three studies of HBsAg^+^ patients receiving antiviral prophylaxis [17,23,35], two studies of HBsAg^+^ patients without prophylaxis [17,23], and ten studies of HBsAg^−^/anti-HBc^+^ patients without prophylaxis [3,17,18,19,20,21,22,23,24,25]. The details of these studies can be found in Table 1. The sample size of the studies ranged from 5 to 81 participants. Of the eleven studies included, nine were conducted in Asian countries [3,17,18,19,21,23,24,25,35] and two in non-Asian countries [20,22]. Regarding the risk of bias, three studies were rated as moderate [18,19,20] and eight as high [3,17,21,22,23,24,25] (Appendix A).

### 3.3. HBV Reactivation Rate in HBsAg^+^ Patients

Three studies including a total of twenty-five participants reported the HBV reactivation rate [17,23,35]. Of these patients, 17 (68%) received antiviral prophylaxis, and none experienced HBV reactivation or hepatitis flare-up (Table 1). Conversely, of the eight HBsAg^+^ patients who did not receive antiviral prophylaxis, 75% (6/8) experienced HBV reactivation. Meta-analysis of the data yielded a pooled reactivation rate of 69.4% (95% CI, 32.9–91.3%) (*I*^2^, 0%; P, 0.38) (Figure 2A). Detailed clinical data for the six HBV reactivation patients are shown in Table 2. All six patients were female, had a median age of 56.5 years (range, 37–78 years), and had a median time from initiation of TCZ treatment to reactivation of 4 months (range, 1–8 months). Of these six patients, three (50%) experienced hepatitis flare-up. All six patients received antiviral treatment, with five receiving entecavir and one receiving adefovir. The antiviral therapy effectively suppressed viral replication and facilitated smooth recovery for these patients. Importantly, no fatalities were recorded in any of these studies.

### 3.4. HBV Reactivation Rate in HBsAg^−^/Anti-HBc^+^ Patients

Ten studies involving 322 patients assessed the HBV reactivation rate in HBsAg^−^/anti-HBc^+^ patients [3,17,18,19,20,21,22,23,24,25]. None of these patients received antiviral prophylaxis. Four patients experienced HBV reactivation, with reported reactivation rates ranging from 0% to 11.1% [3,17,18,19,20,21,22,23,24,25]. Meta-analysis of the data yielded a pooled reactivation rate of 3.3% (95% CI, 1.6–6.7%), and no significant heterogeneity was observed among the included studies (*I*^2^, 0%; P, 0.69) (Figure 2B). Table 2 presented an overview of HBsAg^−^/anti-HBc^+^ patients who experienced HBV reactivation. Among them, 75% were female, with a median age of 67 years (range: 55–75 years). The median time from TCZ treatment initiation to reactivation was 10 months (range: 2–43 months). The reappearance of viral load was observed in three patients, with low fluctuating viral load [19,24]. In two of these patients, the reactivation resolved spontaneously [24]; the third patient achieved successful recovery after receiving antiviral treatment [19]. Seroreversion of HBsAg occurred in only one patient, accompanied by serum HBV DNA levels reaching up to 2.5 × 10^7^ IU/mL without hepatitis flare-up [17]. The patient recovered following antiviral treatment. No deaths were reported.

### 3.5. Subgroup Analysis in HBsAg^−^/Anti-HBc^+^ Patients

Subgroup analysis results are shown in Appendix A. The pooled HBV reactivation rate among anti-HBs^−^ patients was 10.9% (95% CI, 3.5–29.2%; *I*^2^, 0%; P, 0.73) (Figure 2C), and no reactivations occurred in the anti-HBs^+^ group. Anti-HBs^−^ patients had a higher risk of HBV reactivation than did those who were anti-HBs^+^ (Odds ratio, 12.20; 95% CI, 1.16–128.06; *I*^2^, 0%; P, 0.74) (Figure 2D). HBV reactivation rates according to the definition of reactivation were 5.9% (95% CI, 2.4–14.1%; *I*^2^, 0%) among studies using the reappearance or elevation of HBV DNA, 0.6% (95% CI, 0.0–9.0%; *I*^2^, 0%) among those using HBsAg seroreversion, and 1.7% (95% CI, 0.4–6.6%; *I*^2^, 0%) among those considering both criteria. The pooled reactivation rate was 3.7% (95% CI, 1.7–7.8%; *I*^2^, 0%) in Asian populations and 1.7% (95% CI, 0.2–11.1%; *I*^2^, 0%) in non-Asian populations. Subgroup analysis with risk of bias and study design found similar HBV reactivation rates.

### 3.6. Publication Bias and Sensitivity Analysis

The distribution of reports on the risk of HBV reactivation in HBsAg^−^/anti-HBc^+^ patients, as shown in the funnel plot, exhibits a symmetrical pattern, suggesting the absence of publication bias (P, 0.04; Egger’s test) (Appendix A). To evaluate the reliability of the findings, a sensitivity analysis was conducted using the leave-one-out meta-analysis and revealed HBV reactivation rates ranging from 2.2% to 3.8%, with a pooled rate of 3.3% (Appendix A). The consistent results observed for the HBV reactivation rate indicate the robustness of the pooled results.

## 4. Discussion

This systematic review and meta-analysis revealed that TCZ treatment in RA patients leads to immune response changes, potentially triggering HBV reactivation. Among HBsAg^+^ patients, the absence of antiviral prophylaxis was associated with a significant reactivation risk, up to 69.4%. Notably, this risk was effectively eliminated with antiviral prophylaxis. In HBsAg^−^/anti-HBc^+^ patients, the overall reactivation rate of 3.3% was relatively low. However, the median time from TCZ initiation to reactivation was longer compared to the HBsAg^+^ population. Importantly, HBsAg^−^/anti-HBc^+^ patients without anti-HBs demonstrated elevated risk of reactivation compared to anti-HBs^+^ patients.

HBsAg^+^ patients were excluded from previous TCZ clinical trials [12,13], and current guidelines recommend antiviral prophylaxis before immunosuppressive or cytotoxic therapy in this population [14,15,16]. As a result, the reporting of HBV reactivation in HBsAg^+^ patients was limited. Nonetheless, cases of HBV flares, including fatal ones, have been reported in RA patients who are HBsAg^+^ following treatment with tocilizumab [36,37]. In our systematic review, only three studies involving HBsAg^+^ patients receiving TCZ were identified, with a total sample size of 25 individuals [17,23,35]. Among these patients without antiviral prophylaxis, 6 out of 8 experienced HBV reactivation [17,23]. Out of the six cases of HBV reactivation, three had virological reactivations accompanied by hepatitis flares. HBV reactivation was commonly observed in the early stages of treatment, typically occurring within 1–8 months of initiating TCZ therapy. All six patients received antiviral therapy and successfully recovered from their HBV reactivation episodes. In contrast, none of the 17 patients who received prophylaxis experienced reactivation. Therefore, HBsAg^+^ patients treated with TCZ faced a higher risk of HBV reactivation (>10%) [14,15,16], with hepatitis flares observed in 50% of cases. The available evidence emphasizes the significance of antiviral prophylaxis in eliminating this risk and ensuring patient safety [14,15,16].

The HBV reactivation rate was lower in HBsAg^−^/anti-HBc^+^ individuals without antiviral prophylaxis. In a recent meta-analysis of HBsAg^−^/anti-HBc^+^ patients with rheumatoid arthritis receiving IL-6 inhibitor therapy, the pooled reactivation rate was found to be 0.0% (4 studies; 1 reactivation in 162 patients; I^2^, 0%; P, 0.43) [11]. However, our own meta-analysis, which included a greater number of studies and a larger sample size (11 studies; 4 reactivations in 322 patients), demonstrated that this population still has a notable risk of HBV reactivation. Compared to HBsAg^+^ patients, HBsAg^−^/anti-HBc^+^ patients exhibited lower reactivation rates, and the severity of these reactivations was generally mild [17,19,24]. Among the four reported cases of reactivation, three showed fluctuating HBV DNA levels between undetectable and detectable, no HBsAg seroreversion, and normal ALT levels [19,24]. Only one patient experienced a hepatitis flare-up, HBsAg seroreversion, and an elevated serum HBV DNA level of up to 2.5 × 10^7^ IU/mL [17]. The patient recovered from their HBV reactivation episode after receiving antiviral treatment. These findings suggest that HBsAg^−^/anti-HBc^+^ patients still face a moderate risk of HBV reactivation (1–10%) [14,15,16]. Although the reactivations are generally mild, hepatitis flare-ups can still occur.

During the review of HBsAg^−^/anti-HBc^+^ reactivation cases, an interesting finding emerged regarding the potential extended duration of reactivation risk. In contrast to the early reactivation observed in HBsAg^+^ patients (range, 1–8 months) [17,23], HBsAg^−^/anti-HBc^+^ patients exhibited a persistence of reactivation risk lasting up to 43 months after TCZ therapy initiation (range, 2–43 months) [17,19,24]. This observation aligns with a previous study conducted in HBsAg^−^/anti-HBc^+^ patients with rheumatoid arthritis, where the median interval from the initiation of immunosuppressants or bDMARDs to HBsAg seroreversion was 131 months (range, 20–196) and 90 months (range, 10–174), respectively [18]. The potential for a prolonged time interval between reactivation and the initiation of TCZ therapy emphasizes the importance of prolonged monitoring.

Data derived from chemotherapy interventions indicated a higher risk of HBV reactivation in males [38,39]. In contrast, the present study demonstrates a higher propensity for reactivation in females compared to males. Specifically, all cases of reactivation in HBsAg^+^ individuals (6/6) and 75% of cases in HBsAg^−^/anti-HBc^+^ individuals (3/4) were females (Table 2). The result may be linked to the demographic composition of the RA population. Global statistics from the World Health Organization indicate that approximately 70% of individuals diagnosed with RA are female [40]. Our study could not perform further analysis to adjust for other confounding factors related to HBV reactivation. It is difficult to draw a conclusion regarding the sexual effects on HBV reactivation. A recent study with 2845 person-years of follow-up identified factors associated with HBV reactivation in RA patients receiving biologic/targeted synthetic DMARDs, revealing no significant sex differences in the multivariate Cox-regression analysis [41]. Addressing this question necessitates a large-scale cohort study, incorporating an analysis adjusted for potential confounding factors.

The presence of anti-HBs is associated with a reduced risk of reactivation in HBsAg^−^/anti-HBc^+^ patients undergoing chemotherapy for hematological malignancies [26]. This association is also supported by large-scale studies of HBsAg^−^/anti-HBc^+^ patients receiving biologic agents, where the absence of anti-HBs was identified as an independent risk factor for HBV reactivation [18,27,28]. Moreover, a study from Taiwan examined the effect of anti-HBs titer and found that among abatacept-treated patients with rheumatoid arthritis, those with low baseline anti-HBs levels (≤100 mIU/mL) had a higher cumulative risk of HBsAg seroreversion and loss of anti-HBs [18]. However, there was insufficient evidence to recommend the use of anti-HBs for risk stratification in patients receiving TCZ. To address this gap in knowledge, we conducted a subgroup analysis using data from four studies that reported anti-HBs status. The analysis revealed a significantly higher rate of HBV reactivation among anti-HBs^−^ patients, with a pooled rate of 10.9% (95% CI, 3.5–29.2%). In contrast, no instances of reactivation were observed in the anti-HBs^+^ group. Anti-HBs^−^ patients had a significantly higher risk of reactivation than did anti-HBs^+^ individuals (Odds ratio, 12.20; 95% CI, 1.16–128.06). Therefore, anti-HBs may serve as a useful marker for further stratifying the risk of HBV reactivation in HBsAg^−^/anti-HBc^+^ patients receiving TCZ. According to our findings, patients with anti-HBs are at low risk (<1%), while those without anti-HBs are at high risk (>10%) [14,15,16]. Antiviral prophylaxis or regular monitoring should be implemented in patients in the respective risk categories [14,15,16]. Further investigation is required to determine appropriate cut-off values for anti-HBs.

This systematic review and meta-analysis has several limitations. First, most included studies were retrospective, which may lead to irregular monitoring practices and missing data. Single center and retrospective study design lead to the included studies were judged with the moderate-to-high risk of bias, we have to interpret our findings with more cautions before the clinical applications [42]. Incomplete data for levels of HBsAg, hepatitis B e antigen (HBeAg), anti-HBeAb, anti-HBs, and HBV DNA restrict their use for predicting HBV reactivation. Further prospective studies are needed to examine the predictive role of these HBV markers. Second, variations in the definitions used to identify HBV reactivation can result in differences in reported reactivation rates. Although the subgroup analysis conducted on HBsAg^−^/anti-HBc^+^ patients revealed no significant difference in rates based on the definition of HBV reactivation, it remains important to prioritize the use of standardized assessment definitions in future research. Third, the majority of studies included in this review mainly enrolled Asian patients, limiting the generalizability of the findings to patients of other ethnicities. Additional studies from diverse populations are necessary to address this limitation and improve the external validity of the results. Fourth, the potential impact of HBV genotype1 and hepatitis D virus co-infection on reactivation rates were not reported in these including study. Future studies need to address and resolve this issue.

The key strength of this review is the use of updated data from a variety of studies that included patients with RA receiving TCZ therapy. This approach has significantly increased the statistical power, enabling a comprehensive assessment of the risk of HBV reactivation. Furthermore, our meta-analysis fills a gap in previous studies by including subgroup analyses that examine the differential risk of HBV reactivation in patients with and without anti-HBs, revealing the protective effect of anti-HBs. Consequently, this study could provide physicians with evidence-based guidance regarding appropriate antiviral prophylaxis or monitoring strategies based on individual viral status.

## 5. Conclusions

The risk of HBV reactivation from TCZ in RA patients with HBsAg^+^, HBsAg^−^/anti-HBc^+^, and anti-HBs^−^ cannot be ignored but may be avoided. This systematic review highlight the critical importance of the consideration to implement appropriate interventions (e.g., antiviral prophylaxis and routine monitoring policies) for rheumatoid patients to ensure drug safety from TCZ treatment.

## Figures and Tables

**Figure 1 viruses-16-00078-f001:**
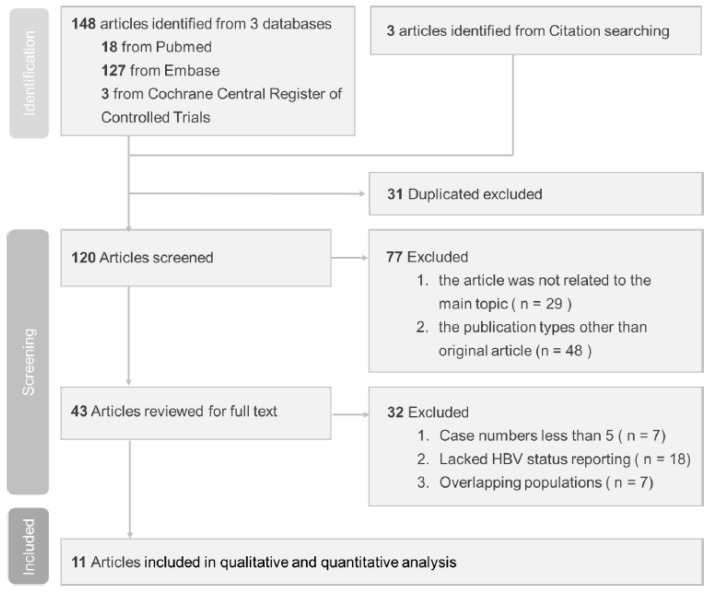
Study selection based on PRISMA diagram.

**Figure 2 viruses-16-00078-f002:**
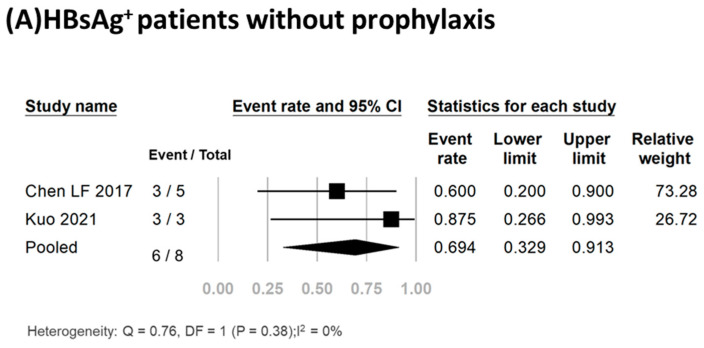
Pooled rates of HBV reactivation [3,17,18,19,20,21,22,23,24,25,26].

**Table 1 viruses-16-00078-t001:** Demographic data and characteristics of cohorts of included studies.

**1. HBsAg^+^ Patients**													
**(A) With Prophylactics (n = 17)**													
**First Author** **(Year)**	**Location**	**Setting/** **City**	**Study Design**	**Age** **(Years)**	**TCZ** **Dose/Duration**	**Follow-Up**	**Definition of HBVr**	**Prophylactic**				**Definition of Hepatitis** **Flare-Up**	**HBVr** **Associated Hepatitis/Death**
								**Total (n)**		**HBVr (n)**			
Chen LF 2017 [23]	China	S/Guangzhou	P	46 #	IV 8 mg/kg Q4 weeks for 12 weeks	12 weeks	HBV DNA reappearance or elevation	2		0		AST or ALT elevated >2 ULN	0/0
Lin 2019 [35]	Taiwan	S/Taipei	R	59 *	IV 4 mg/kg Q4 weeks than 4–8 mg/kg Q4 weeks	3 years	NR	11		0		NR	0/0
Kuo 2021 [17]	Taiwan	S/Chiayi	R	65 #	IV 4–8 mg/kg Q4 weeks or SC 162 mg Q1–2 weeks	9 years	HBV DNA reappearance or elevation	4		0		ALT elevated 3 times and >100 U/L	0/0
**(B) Without Prophylactics (n = 8)**													
**First Author** **(Year)**	**Location**	**Setting/** **City**	**Study Design**	**Age** **(Years)**	**TCZ** **Dose/Duration**	**Follow-Up**	**Definition of HBVr**	**Non-Prophylactic**			**HBVr Time (Median; Range)**	**Definition of Hepatitis** **Flare-Up**	**HBVr** **Associated Hepatitis/Death**
								**Total (n)**		**HBVr (n)**			
Chen LF 2017 [23]	China	S/Guangzhou	P	46 #	IV 8 mg/kg Q4 weeks/ 12 weeks	12 weeks	HBV DNA reappearance or elevation	5		3	3 (1–3) months	AST or ALT elevated >2 ULN	0/0
Kuo 2021 [17]	Taiwan	S/Chiayi	R	65 #	IV 4–8 mg/kg Q4 weeks or SC 162 mg Q1–2 weeks	9 years	HBV DNA reappearance or elevation	3		3	6 (5–8) months	ALT elevated 3 times and >100 U/L	3/0
**2. HBsAg^−^/Anti-HBc^+^ Patients without NA Prophylaxis (n = 322)**													
**First Author (Year)**	**Location**	**Setting/** **City**	**Study Design**	**Age** **(Years)**	**TCZ** **Dose/Duration**	**Follow-Up**	**Definition of HBVr**	**Patients with HBVr, (n/N)**	**Anti-HBs^+^** **(n)**	**Anti-HBs^−^** **(n)**	**HBVr Time (Median; Range)**	**Definition of Hepatitis** **Flare-Up**	**HBVr Associated Hepatitis/Death**
Mori 2011 [25]	Japan	S/Kumamoto	R	73 # (overall)	NR/6 months	NR	HBV DNA reappearance or elevation	0/5	NR	NR	NIL	NR	0/0
Nakamura 2016 [24]	Japan	S/Tochigi	R	64 # (overall)	NR/ 18 months	1.5 years	HBV DNA reappearance or elevation	2/18	NR	NR	All 2 months	NR	0/0
Chen LF 2017 [23]	China	S/Guangzhou	P	46 #	IV 8 mg/kg Q4 weeks/12 weeks	12 weeks	(1) HBV DNA reappearance or elevation, or (2) HBsAg seroreversion	0/41	32	9	NIL	AST or ALT elevated >2 ULN	0/0
Ahn 2018 [3]	korea	S/Seoul	R	57 #	NR/10.8 months	9.4 years	HBV DNA reappearance or elevation	0/15	12	3	NIL	NR	0/0
Papalopoulos 2018 [22]	Greece	S/Heraklion	R	65 # (overall)	NR/NR	2 years	HBV DNA reappearance or elevation	0/30	NR	NR	NIL	NR	0/0
Tien 2018 [21]	Taiwan	S/Changhua	P	56.3 * (overall)	NR/NR	3.5 years	(1) HBV DNA reappearance or elevation, or (2) HBsAg seroreversion	0/16	NR	NR	NIL	NR	0/0
Carlino 2019 [20]	Italy	M/9 Apulian rheumatologic centres	R	57 *	NR/NR	8.3 years	NR	0/27	NR	NR	NIL	NR	0/0
Watanabe 2019 [19]	Japan	S/Sapporo	R	68 #(overall)	NR/NR	1.3 years	HBV DNA reappearance or elevation	1/25	21	4	43 months	NR	0/0
Chen MH 2021 [18]	Taiwan	S/Taipei	R	51.8 *	NR/NR	16 years	HBsAg seroreversion	0/81	NR	NR	NIL	ALT elevated >2 ULN and >120 U/L	0/0
Kuo 2021 [17]	Taiwan	S/Chiayi	R	67 #	IV 4–8 mg/kg Q4 weeks or SC 162 mg Q1–2 weeks	9 years	HBV DNA reappearance or elevation	1/64	37	14	18 months	ALT elevated 3 times and >100 U/L	1/0

* median; # mean. ALT, alanine aminotransferase; AST, aspartate aminotransferase; HBV, hepatitis B virus; HBVr, HBV reactivation; HBeAg, hepatitis B e antigen; HBsAg, hepatitis B surface antigen; M, multi-center; NR, data not reported; NIL, do not need the data; ULN, upper limited normal; P, prospective; RA, Rheumatoid arthritis; R, retrospective; S, single-center; SC, Subcutaneous; TCZ, tocilizumab.

**Table 2 viruses-16-00078-t002:** Characteristics and outcomes of rheumatoid arthritis patients receiving tocilizumab who experienced HBV reactivation.

HBsAg^+^ Patients										
Case	Age/Sex	DMARDs	Reactivation Time	HBV DNA at Peak (IU/mL)	ALT (Baseline/Peak; IU/L)	Antiviral Therapy	Hepatitis Flare-Up		Outcome	Reference
1	43/F	MTX/HCQ/SSZ	3 months	8.17 × 10^4^	26/30	Adefovir	N		living	L.F. Chen 2017 [23]
2	37/F	HCQ	3 months	500	11/20	Entecavir	N		living	L.F. Chen 2017 [23]
3	57/F	MTX/HCQ/SSZ	1 month	500	13/19	Entecavir	N		living	L.F. Chen 2017 [23]
4	56/F	MTX/Pd/SSZ/CSA	5 months	3.7 × 10^7^	155/155	Entecavir	Y		living	M.H. Kuo 2021 [17]
5	77/F	MTX/Pd/SSZ/LEF/CSA	6 months	1.6 × 10^7^	698/946	Entecavir	Y		living	M.H. Kuo 2021 [17]
6	78/F	MTX/Pd/SSZ/LEF	8 months	1.7 × 10^8^	106/355	Entecavir	Y		living	M.H. Kuo 2021 [17]
**HBsAg^−^/anti-HBc+ Patients**										
**Case**	**Age/Sex**	**DMARDs**	**Reactivation Time**	**HBV DNA at Peak (IU/mL)**	**ALT (Baseline/Peak; IU/L)**	**Antiviral Therapy**	**Hepatitis Flare-Up**	**Sero-Conversion**	**Outcome**	**Reference**
1	75/M	MTX	2 months	fluctuated between being undetectable or detectable	NR/NR	NR	NR	NR	living	J. Nakamura 2016 [24]
2	55/F	MTX/Pd	2 months	<2.1 log copies/mL	NR/NR	NR	NR	NR	living	J. Nakamura 2016 [24]
3	67/F	MTX/Pd	43 months	<20	10/15	NAA	NR	NR	living	T. Watanabe 2018 [19]
4	67/F	MTX/Pd/SSZ/LEF	18 months	2.5 × 10^7^	98/107	Entecavir	N	Y	living	M.H. Kuo 2021 [17]

ALT, alanine aminotransferase; DMARDs, disease-modifying anti-rheumatic drugs; HBsAg, HBV surface antigen; HBVr, HBV reactivation; anti-HBc, hepatitis B core antibody; HCQ, hydroxychloroquine; LEF, leflunomide; MTX, methotrexate; SSZ, sulfasalazine; CSA, cyclosporine; Pd, prednisolone; LEF, leflunomide; NAA, nucleic acid analog.

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
