# Peer review of "The Risk of Hepatitis B Virus Reactivation in Rheumatoid Arthritis Patients Receiving Tocilizumab: A Systematic Review and Meta-Analysis"

_viruses, 2024, doi:10.3390/v16010078_

Round 1

Reviewer 1 Report

Comments and Suggestions for Authors

The manuscript “The Risk of Hepatitis B Virus Reactivation in Rheumatoid Arthritis Patients Receiving Tocilizumab: A Systematic Review and Meta-Analysis” presents interesting and valuable information about risks of HBV infection reactivation in patients receiving immunomodulatory drug for rheumatoid arthritis. There are some points to be considered:

For easier following the text, I suggest using anti-HBc and anti-HBs instead of HBcAb and HBsAb throughout the text, tables and figures.

The levels of reactivation risks (low-below 1%, moderate-below 10%, high-above 10%) should be explained in the Discussion before line 256, where it is stated that HBsAg-positive patients have a higher risk for reactivation.

In Table 2, information about seroconversion of one patient to HBsAg positive should be added.

Comments on the Quality of English Language

Minor editing of English language required.

Reviewer 2 Report

Comments and Suggestions for Authors

In this Systemic Review by Ko and colleagues, the authors perform a meta-analysis of the literature on HBV reactivation following Tocilizumab (Toc; an IL-6 inhibitor) treatment for rheumatoid arthritis (RA). Toc clinical trials have largely excluded HBV patients. Hence, the extent to which this drug can lead to HBV reactivation is largely unknown, and previous studies to address this have been somewhat contradictory. Here, the author's meta-analysis indicates that Toc therapy modestly increases the risk for HBV reactivation, indicating that antiviral prophylaxis should be considered when Toc is given to RA patients with a history of HBV infection. The analysis appears rigorous, and the findings reveal some new biological and clinical relevance regarding IL-6 and HBV. This reviewer does not have specific expertise in the statistical methods of meta-analyses, so my comments come from the viewpoint of a general scientific audience.

1. The authors’ literature search identified 148 articles that were trimmed to 11 for analysis. While the selection criteria are thoughtfully considered, it is unclear why 7 studies were excluded due to a case number of < 5. This seems arbitrary. What is the statistical justification for including a study with 5 patients but not 4?

2. The analysis employs 10 studies examining HBV reactivation rates in HBsAg-/HBcAb+ patients. The frequency of reactivation in the studies ranged from 0-11.1%. While the authors’ finding of a pooled reactivation rate of 3.3% (95% CI of 1.6-6.7%) seems reasonable, this would also appear mostly predictable.

3. It seems that the risk of reactivation is much greater in females than males (100% of HBsAg+ and 75% of HBsAg-/HBcAb+ reactivations were female). However, it is unclear whether this is reflective of the underlying study population (as RA affects more women than men) or is somehow biologically relevant to HBV or Toc. The authors should comment further on this finding.

4. As part of their analyses, the authors perform a “quality assessment of the included cohort studies” using something called the Newcastle-Ottawa Scale to estimate the risk of bias. The results of this analysis show that 8 of the underlying studies have a high risk of bias, 3 were moderate, and none were low. The implications of this are unclear. Is this typical? The authors should discuss this in further detail.

5. This is probably not possible due to the limitations of the information provided in the underlying studies, but it would be interesting to know if the reactivation rates were impacted by either HBV genotype or HDV co-infection.

6. This is perhaps a computer glitch of some sort, but Suppl Table 1 appears to be missing in the file that I downloaded.

Comments on the Quality of English Language

Minor editing is required.
